# Metabolic signature in nucleus accumbens for anti-depressant-like effects of acetyl-L-carnitine

Antoine Cherix[1]*, Thomas Larrieu[2†], Jocelyn Grosse[2], João Rodrigues[2], Bruce McEwen[3], Carla Nasca[3], Rolf Gruetter[1], Carmen Sandi[2]*

[1]Laboratory for Functional and Metabolic Imaging (LIFMET), École Polytechnique Fédérale de Lausanne (EPFL), Lausanne, Switzerland; [2]Laboratory of Behavioral Genetics, Brain and Mind Institute, School of Life Sciences, Ecole Polytechnique Fédérale de Lausanne, Lausanne, Switzerland; [3]Harold and Margaret Milliken Hatch Laboratory of Neuroendocrinology, The Rockefeller University, New York, United States

**\*For correspondence:**
antoine.cherix@epfl.ch (AC);
carmen.sandi@epfl.ch (CS)

**Present address:** [†]Center for Psychiatric Neurosciences, Lausanne University Hospital (CHUV), Site de Cery, Lausanne, Switzerland

**Competing interests:** The authors declare that no competing interests exist.

**Abstract** Emerging evidence suggests that hierarchical status provides vulnerability to develop stress-induced depression. Energy metabolic changes in the nucleus accumbens (NAc) were recently related to hierarchical status and vulnerability to develop depression-like behavior. Acetyl-L-carnitine (LAC), a mitochondria-boosting supplement, has shown promising antidepressant-like effects opening therapeutic opportunities for restoring energy balance in depressed patients. We investigated the metabolic impact in the NAc of antidepressant LAC treatment in chronically-stressed mice using ¹H-magnetic resonance spectroscopy (¹H-MRS). High rank, but not low rank, mice, as assessed with the tube test, showed behavioral vulnerability to stress, supporting a higher susceptibility of high social rank mice to develop depressive-like behaviors. High rank mice also showed reduced levels of several energy-related metabolites in the NAc that were counteracted by LAC treatment. Therefore, we reveal a metabolic signature in the NAc for antidepressant-like effects of LAC in vulnerable mice characterized by restoration of stress-induced neuroenergetics alterations and lipid function.

## Introduction

Depression is among the leading causes of disability worldwide, which reflects the current lack of understanding of its underlying mechanisms (*Friedrich, 2017*; *Menke, 2018*). Metabolic alterations are emerging as key etiological factors for the development of neuropsychiatric disorders, including depression (*Pei and Wallace, 2018*; *Andreazza and Nierenberg, 2018*; *Kim et al., 2019*). The strong reliance of the brain on high energy consumption would make it particularly vulnerable to metabolic alterations (*Pei and Wallace, 2018*). In addition, chronic stress has a strong capacity to trigger and exacerbate depression (*de Kloet et al., 2005*; *Richter-Levin and Xu, 2018*) and impinges metabolic-costly neuronal adaptations in structure and function (*Turner and Lloyd, 2004*; *de Kloet et al., 2005*; *McEwen et al., 2015*). Accordingly, stress-associated depletion of brain's energy resources could lead to impaired neuronal plasticity underlying depression (*Morava and Kozicz, 2013*; *Picard et al., 2018*). Mitochondria, by powering the brain with energy production, play a central role in the adaptation and response to stress (*Picard et al., 2015*), and mitochondrial supplements could provide an efficient means of protecting brain structures that are particularly vulnerable to stress (*Parikh et al., 2009*).

However, not all individuals are equally affected by stress (*Duclot and Kabbaj, 2013*; *McEwen et al., 2015*; *Russo et al., 2012*); while some individuals show a high vulnerability to

develop depression, others endure resilience following stress exposure (*Russo et al., 2012*; *Weger and Sandi, 2018*). It remains unclear which factors provide resilience to stress in certain individuals and what are the underlying mechanisms (*Larrieu and Sandi, 2018*; *Ménard et al., 2017*). In addition to the great predictive power of high anxiety trait in defining stress vulnerability (*Sandi et al., 2008*; *Castro et al., 2012*; for reviews, see *Sandi and Richter-Levin (2009)*; *Russo et al., 2012*; *Weger and Sandi, 2018*), epidemiological, clinical and animal work point to a link between social hierarchies and depression (*Larrieu and Sandi, 2018*).

Recently, in the C57BL/6J inbred mouse strain, we found that high rank animals were more susceptible to display social avoidance following exposure to chronic social defeat stress (CSDS), while low rank mice were not affected (*Larrieu et al., 2017*). Data from [1]H-magnetic resonance spectroscopy [[1]H-MRS; one of the few non-invasive methods that can provide direct information on brain metabolism in vivo (*Duarte et al., 2012*)] revealed a relationship between the metabolic profile of the nucleus accumbens [NAc; a hub brain region for the regulation of motivated behaviors (*Robbins and Everitt, 1996*) implicated in the pathophysiology of depression (*Francis and Lobo, 2017*)], social rank, and vulnerability to stress. Thus, while under basal conditions low rank showed lower levels of energy-related metabolites than high rank mice, it was only the low rank/resilient group that displayed increased metabolite levels following CSDS (*Larrieu et al., 2017*). These observations suggested that metabolic targeting may be an optimal treatment intervention and confirmed that NAc is a particularly sensitive structure that might beneficiate from energetic support.

Acetyl-L-carnitine (LAC) has been recently shown to have promising potency to rapidly alleviate depressive-like symptoms in preclinical studies (*Bigio et al., 2016*; *Lau et al., 2017*; *Wang et al., 2015*; *Nasca et al., 2013*) and in humans, where emerging clinical evidence supports its good tolerability (*Wang et al., 2014*; *Veronese et al., 2018*). LAC is an endogenous short-chain acetyl ester of free carnitine involved in the transport of long chain fatty acids into the mitochondria for degradation by beta oxidation thus, contributing to energy metabolism (*Ferreira and McKenna, 2017*). In addition, LAC can facilitate the removal of oxidative products, provide acetyl groups for protein acetylation, be used as a precursor for acetylcholine, or be incorporated into neurotransmitters such as glutamate, glutamine and GABA (*Ferreira and McKenna, 2017*). However, it is not known whether LAC treatment can counteract brain metabolic alterations specifically observed in the context of stress-induced depression.

In this study, we investigated the ability of LAC supplementation to protect vulnerable mice against stress induced depressive-like behaviors. As indicated above, socially dominant C57BL/6J mice, were at higher risk of developing depression-like behavior following exposure to CSDS (*Larrieu et al., 2017*). Here, in order to exclude that the identified vulnerability is a mere reflect of the social stressor used (*Larrieu and Sandi, 2018*), a first aim of this study was to assess the link between social rank and vulnerability to develop depressive-like behaviors using chronic exposure to a non-social (e.g., physical) stressor. To this end, and given that lipid peroxidation has been shown to be increased by restrain stress in the striatum (*Atif et al., 2008*), we exposed mice to the well-established 21 day restrain stress protocol (*Lau et al., 2017*; *Nasca et al., 2015*). Subsequently, we studied the impact of LAC treatment coinciding with the last week of stress exposure on the concentration of up to 20 metabolites in the NAc using in vivo [1]H-MRS at 14 Tesla (*Larrieu et al., 2017*; *Duarte et al., 2012*; *Mlynárik et al., 2008*). We also tested mice for depressive-like behaviors, including motivation to explore social conspecifics and coping responses in the forced swim test (FST) (*Nestler et al., 2002*). Besides the evident translational potential of [1]H-MRS at 14 T in identifying biomarkers, rodent [1]H-MRS studies at ultra-high field bridge a potential pathological indicator and its associated molecular signature to physiological mechanisms.

## Results

### High rank mice are vulnerable to chronic restraint stress

First, we confirmed our initial observations (*Larrieu et al., 2017*) regarding the behavioral phenotype of high and low rank mice (as assessed by testing the four mice from the same home cage in the social confrontation tube test; SCTT), under basal conditions (i.e., before any stress was applied). Specifically, we report here that high (ranks 1 and 2) and low (ranks 3 and 4) rank mice (*Figure 1B and C*; time in the tube 1–2 vs 1–4, p<0.001), displayed different profile for anxiety-like behaviors. It

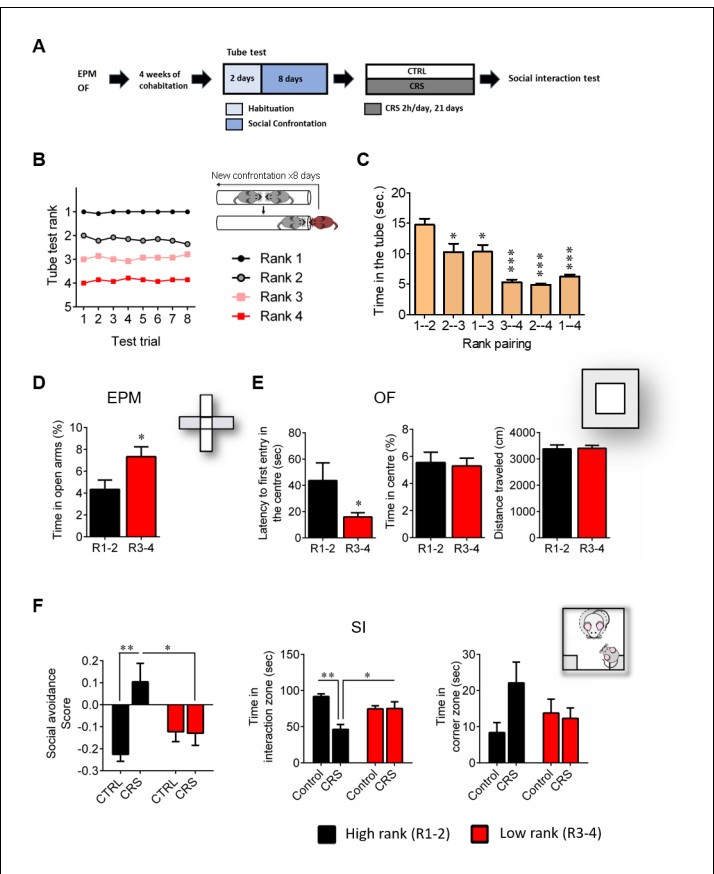

**Figure 1.** High rank mice exhibit susceptible behavioral phenotype after 21 days of chronic restraint stress. (A) Experimental design of the restraint stress protocol. (B) Summary of nine cages representing the SCTT ranks and winning times as a function of SCTT trials over the 8 days of test. (C) Time spent in tube as a function of rank pairing ($F_{5,35}=18.19$, p<0.0001, one-way ANOVA; *p<0.05, ***p<0.001, Bonferroni's test, n = 7 per rank pairing). (D) Anxiety-like behaviors measured as the percent time spent in the open arms of an elevated plus maze after segregation into high rank vs low rank mice (p*<0.05, unpaired t-test, two-tailed, n = 14 per group). (E) Anxiety-related behaviors measured in the open-field, including latency to first enter the center of the arena (*p<0.05, unpaired t-test, two-tailed n = 14 per group) and time in center zone (n.s. unpaired t-test, two-tailed n = 14 per group). Locomotor activity is measured as the distance travelled in the OF (n.s. unpaired t-test, two-tailed n = 14 per group). (F) Social interaction (SI) test measured after chronic restraint stress protocol in high rank vs low rank animals (Social avoidance score: Interaction: $F_{1,21}=7.75$, p<0.05; rank effect: $F_{1,21}=1.18$, p>0.05; stress effect: $F_{1,21}=7.15$, p<0.05, two-way ANOVA; p*<0.05, **p<0.01, Bonferroni's test, n = 6–7 per group/Time in interaction zone: Interaction: $F_{1,21}=12.80$, p<0.005; rank effect: $F_{1,21}=0.85$, p<0.05; stress effect: $F_{1,21}=12.21$, p<0.005, two-way ANOVA; p*<0.05, **p<0.005, Bonferroni's test, n = 6–7 per group/Time in corner zone: Interaction: $F_{1,21}=3.29$, p>0.05; rank effect: $F_{1,21}=0.28$, p<0.05; stress effect: $F_{1,21}=2.14$, p>0.05, two-way ANOVA, n = 6–7 per group). Data are displayed as mean ± SEM.

was reflected by a higher time spent in the open arms of an elevated plus maze (EPM) (*Figure 1D*; p<0.05) as well as an increased latency to enter the center of an open field (OF) (*Figure 1E*, left; p<0.05) in high rank compared to low rank mice. However, the two groups did not show differences in the percent time spent in the center of the OF (*Figure 1E*, center), indicating that their difference in anxiety-like behaviors depends on the specific threat encountered. Importantly, no difference in locomotor activity was observed, as the distance travelled by both groups in the OF was similar (*Figure 1E*, right; n.s.).

Strikingly, we revealed that the susceptibility to CSDS observed in high rank mice in our previous study (*Larrieu et al., 2017*) can be generalized to a non-social, chronic restraint stress (CRS) protocol. Indeed, high rank individuals were the ones that, after CRS, showed vulnerability to develop social avoidance towards an unfamiliar mouse in a social interaction test (SI) (*Figure 1F*). In contrast,

low rank mice seemed not being affected by stress exposure (social avoidance score: interaction, $F_{1,21}=7.75$, p<0.05; time in interaction zone: interaction, $F_{1,21}=12.80$, p<0.005).

## LAC treatment partially abolishes stress-induced behavioral vulnerability in high rank mice

Given the emerging evidence indicating a potential therapeutic efficiency of LAC, the acetylated form of carnitine, in the context of depression (see Introduction), we tested whether LAC treatment could counteract the induction of depressive-like behaviors by CRS in vulnerable (i.e., high rank) mice. In this experiment, mice were exposed to the CRS protocol and received concomitant administration of LAC during the last 7 days of the stress period. Animals exposed to CRS displayed a significant decrease in cumulative body weight gain, regardless of their social rank, that was not counteracted by LAC supplementation (*Figure 2B*; in high rank: Stress effect $F_{2,10} = 45.0$, p<0.0001 and *Figure 2—figure supplement 1A*; in low rank: Stress effect $F_{2,10} = 20.4$, p<0.001). Importantly, whereas stress led to an increase in liquid consumption (+15 ± 12%) in the stressed groups (*Figure 2C* and *Figure 2—figure supplement 1B*), there was no difference in liquid consumption between the CRS and CRS + LAC groups (*Figure 2C*).

As we showed in our previous and current studies that CSDS and CRS induce social avoidance only in high rank mice, we investigated whether LAC supplementation could attenuate the effects of chronic stress on emotional behavior in this susceptible (i.e., high rank) group. We found that LAC treatment did not prevent the increase in social avoidance induced CRS in a group of 6 high rank mice (*Figure 2D*; n.s.). Since LAC has been shown to be a mitochondria-boosting supplement, we tested the possibility that LAC would be more efficient in a high energy-demanding test such as the FST. Indeed, LAC treatment was effective to abolish CRS-induced increase in passive coping behaviors in high rank mice in the FST (*Figure 2E*). Specifically, a significant increase in the immobility time observed in high rank mice following CRS exposure (*Figure 2E*; $F_{2,15}=5.31$, p<0.05), was prevented by LAC supplementation (p<0.05). In order to obtain an integrated estimation of how stress and LAC treatments affect emotional behavior more globally, we computed an overall behavioral composite score to integrate deviation from normality considering high rank mice variance in both behavioral tests. Specifically, the use of this composite score allows considering variation from the mean in individuals' behavior across two different behavioral tests, providing a more robust measurement of individuals' behavior in tests that are typically used to index mice depressive-like behaviors. ANOVA of these data indicated a significant effect of treatments (*Figure 2F*; $F_{2,15}=10.31$, p<0.005); specifically, CRS led to increased depressive-like behaviors (p<0.005) that was restored by LAC (p<0.05). Altogether, our findings support the view that pharmacological enhancement of mitochondrial function by LAC supplementation normalizes behavioral changes in stressed-high rank mice under unescapable adversity.

## [1]H-MRS in NAc reveals stress-responsive metabolites in high rank mice counteracted by LAC treatment

Using in vivo [1]H-MRS, we aimed at revealing the NAc neurochemical and metabolite profile in high rank mice following CRS and LAC supplementation. Our [1]H-MRS acquisitions led to a spectral signal-to-noise ratio (SNR) of 17.5 ± 0.3 with a linewidth of 16 ± 1 Hz after shimming with FAST(EST) MAP. The acquired NAc spectra allowed us to quantify up to 20 metabolites with LCModel (*Figure 3*). First, we applied an unbiased multivariate factor analysis (FA) that revealed three main factors that accounted for 31%, 12% and 9% of total variance (*Figure 4*). Individual metabolites with loadings above 0.4 in Factor one included taurine (Tau), glutamate (Glu), phosphocreatine (PCr), N-acetylaspartate (NAA), γ-aminobutyric acid (GABA), creatine (Cr), myo-inositol (Ins), aspartate (Asp), phosphocholine (PCho), glucose (Glc), glutathione (GSH) and ascorbate (Asc) (*Figure 5*). Metabolites with loading above 0.4 for the two other components were PCho, glycerophosphorylcholine (GPC), and Asc for Factor two, and glutamine (Gln) and lactate (Lac) for Factor three (*Figure 6*).

Only Factor one was able to discriminate for stress and treatment response between high rank/vulnerable mice (*Figure 4B*; $F_{2,13}=7.04$; p<0.01). Specifically, in high rank mice, CRS led to a reduction in factor one metabolites as compared to their non-stressed high rank counterparts (p<0.005). Importantly, LAC treatment reversed the effect of CRS on Factor one metabolic profile (p<0.05). Interestingly, we further found a significant negative correlation between the behavioral composite

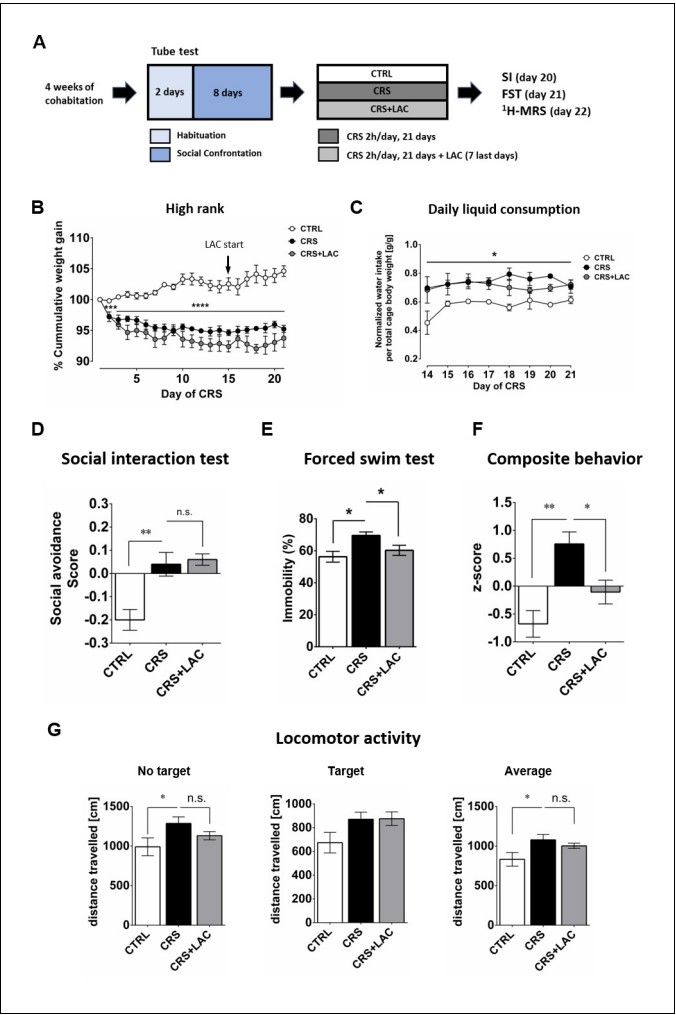

**Figure 2.** High rank mice respond to acetyl-L-carnitine treatment after chronic restraint stress. (**A**) Experimental design of the restraint stress protocol and treatment procedure. (**B**) High rank mice show a reduction of cumulative weight gain during the restraint stress protocol (Interaction: $F_{20,40}$=11.5, p<0.0001; stress effect: $F_{2,10}$ = 45.0, p<0.0001, repeated measures two-way ANOVA; ***p<0.001, ****p<0.0001, Bonferroni's test, n = 6 per group). The start of LAC treatment during CRS protocol is indicated with an arrow (day 14). (**C**) Daily water intake during the LAC treatment period (given during the last week of the CRS protocol) normalized by total body weight of the four mice per cage (Group effect: $F_{2,4}$=17.0, *p<0.05; Interaction: $F_{14,28}$=0.90, p>0.05; time effect: $F_{7,14}$=1.24, p>0.05, repeated measures two-way ANOVA; n = 3 cages per group). Thus, water intake data represent the cage average value. Liquid consumption during the first days of the CRS protocol is shown in *Figure 2—figure supplement 1B* (**D**) Social avoidance scores measured after chronic restraint stress protocol in high rank mice ($F_{2,15}$=12.08, p<0.01, one-way ANOVA; **p<0.01, Bonferroni's test, n = 6 per group). (**E**) Behavioral despair measured with a forced swim test between high rank mice ($F_{2,15}$=5.31, p<0.05, one-way ANOVA; *p<0.05, Bonferroni's test, n = 6 per group). (**F**) Depressive-like behavior measured as a composite z-score component of social avoidance and immobility time between high rank mice ($F_{2,15}$=10.31, p<0.005, one-way ANOVA; *p<0.05**p<0.005, Bonferroni's test, n = 6 per group). (**G**) Locomotor activity measured during the SI test (No target present: $F_{2,15}$=2.94, p>0.05, one-way ANOVA; *p<0.05, Bonferroni's test/Target present: $F_{2,15}$=2.79, p>0.05, one-way ANOVA;/Average: $F_{2,15}$=3.67, p<0.05, one-way ANOVA; *p<0.05, Bonferroni's test, n = 6 per group). Effect of LAC on low rank is shown in *Figure 2—figure supplement 1*.

The online version of this article includes the following figure supplement(s) for figure 2:

**Figure supplement 1.** Physiological and behavioral readouts following CRS and LAC treatment in low rank mice.

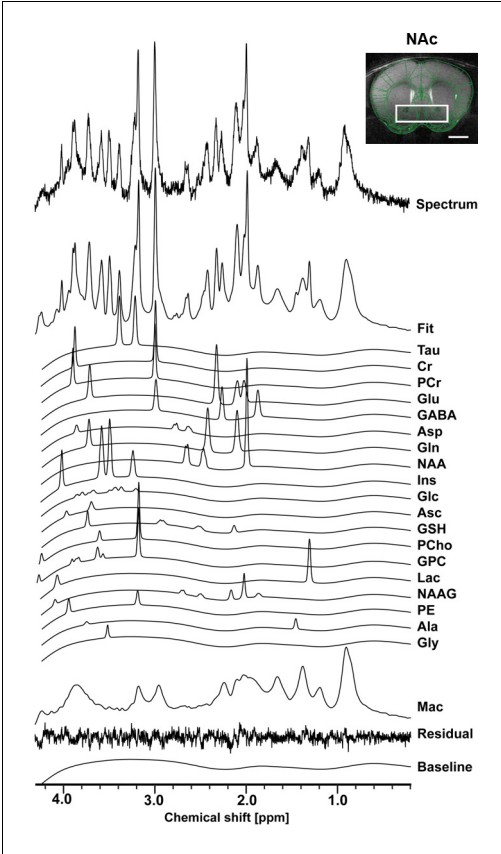

**Figure 3.** The neurochemical profile of the nucleus accumbens measured with in vivo [1]H-MRS at 14T. Spectrum fitting and neuroanatomical image of the NAc with respective voxel position in mouse brain. Spectrum is decomposed into the total fit, the individual metabolite components of the fit, the residual and the baseline, as a result of LCModel analysis. The fitted neurochemical profile included following metabolites: taurine (Tau), creatine (Cr), phosphocreatine (PCr), glutamate (Glu), γ-aminobutyric acid (GABA), aspartate (Asp), glutamine (Gln), N-acetyl-aspartate (NAA), myo-inositol (Ins), glucose (Glc), ascorbate (Asc), glutathione (GSH), phosphorylcholine (PCho), glycerophosphorylcholine (GPC), lactate (Lac), N-acetylaspartyl-glutamate (NAAG), phosphoethanolamine (PE), alanine (Ala), glycine (Gly), as well as macromolecules (Mac).

z-score for depressive-like behaviors in high rank mice and their Factor one metabolite levels (*Figure 4C*; R = −0.58, p<0.05). This finding supports a link between NAc metabolic profile and depression-like behaviors in stress-vulnerable and LAC-treated mice.

Then, we selected the metabolites from Factor one with loadings above 0.4 to perform specific analyses to assess differences between the three groups (i.e., controls, CRS and CRS + LAC) of high rank mice. In metabolites strongly loading in factor one, we found that the observed stress effect was mainly carried by changes in Tau (p<0.05), PCr (p<0.05), Glu (p<0.05), Asp (p<0.05) and NAA (p<0.05), while the stress-reversing effects of LAC were driven by Tau (p<0.05) and PCr (p<0.05) (*Figure 5*). Among the rest of metabolites that either loaded below 0.5 in Factor one or loaded in the other two factors (*Figure 6A*), we found that Gln and PCho were similarly reduced by CRS (p<0.05) and reversed by LAC treatment (p<0.05). Notably, the ratio of GPC over PCho, that is the level of degradation product of phospholipids over their precursor, respectively, was reduced by LAC (p<0.05). In further correlational analyses addressed to identify potentially relevant treatment targets, we found two negative correlations, one between immobility time in the FST and taurine levels (R = −0.52, p<0.05) and a second one between the composite behavioral z-score and GABA (R = −0.55, p<0.05) (*Figure 6B–D*).

## Discussion

In this study, using in vivo [1]H-MRS at 14 T, we identified a chronic stress-related metabolic signature in the NAc of high rank mice -identified as a group of high vulnerability to develop depressive-like behaviors- that was partially restored by LAC treatment. High rank animals were defined as those that following cohabitation in groups of 4 males, emerged as ranks 1 and 2 in the social confrontation tube test. Our results provide an in vivo metabolic basis for understanding of antidepressant-like properties of LAC and its protective effects against chronic stress.

When exposed to chronic social defeat (i.e., daily exposure and defeat by an aggressive mouse), high rank C57BL/6J mice living in tetrads are at higher risk of developing social avoidance (i.e., a depressive-like behavior) than low rank mice (*Larrieu et al., 2017*). Our results reinforce the view that social rank in mice predicts vulnerability to stress. In addition, given the non-social character of the CRS protocol, our results argue against the view that their vulnerability would be solely based upon loss of social status (*Larrieu and Sandi, 2018*). It is important to note that, under basal conditions (i.e., before stress application), higher rank is related to higher anxiety-like behaviors as indicated by their exploration of an elevated plus maze (but note that these animals did not differ from low rank in the time spent in the center of the open field, only in their latency to enter the zone), as

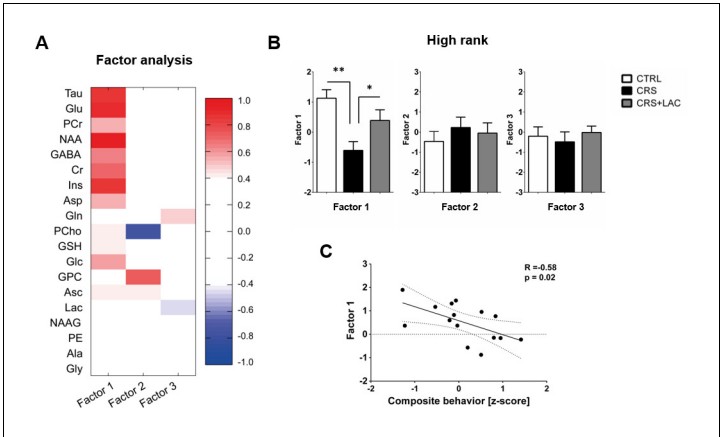

**Figure 4.** Factor analysis identified one main factor that accounts for treatment-related effects in the metabolic profile of nucleus accumbens in high rank mice. (**A**) Metabolites in the nucleus accumbens that load into Factor one, Factor two and Factor three of the factor analysis. The heat map represents the individual loadings of each metabolite into each factor. Factor one represents a linear combination that summarizes neurochemical changes including metabolites with strong contribution (above 0.5: Tau, Cr, PCr, Glc, Glu, GABA, Asp, NAA and Ins.) and moderate (0.4–0.5: GSH and Asc). (**B**) CRS and LAC treatment in CRS-treated high rank mice impact on Factor one metabolites ($F_{2,13}$=7.04, p<0.01, one-way ANOVA; *p<0.05, **p<0.01, Fisher LSD test n = 5–6 per group) (**C**) Factor one correlates with the composite emotional (i.e., depressive-like) behavior in high rank animals (R = −0.58; p<0.05).

high anxiety trait is a well-established risk factor to develop stress-related depressive behaviors (*Sandi et al., 2008*; *Castro et al., 2012*; for reviews, see *Sandi and Richter-Levin (2009)*; *Russo et al., 2012*; *Weger and Sandi, 2018*). However, a note of caution should be added as previous work suggests that whereas dominant mice tend to display more novelty-related exploratory behavior than less dominant mice, differences in anxiety-related behaviors seem to be less consistent (see *Varholick et al., 2018*, and references herein). We would also like to emphasize that we find a high reliability and linearity of social rank assessed under our experimental conditions that involve at

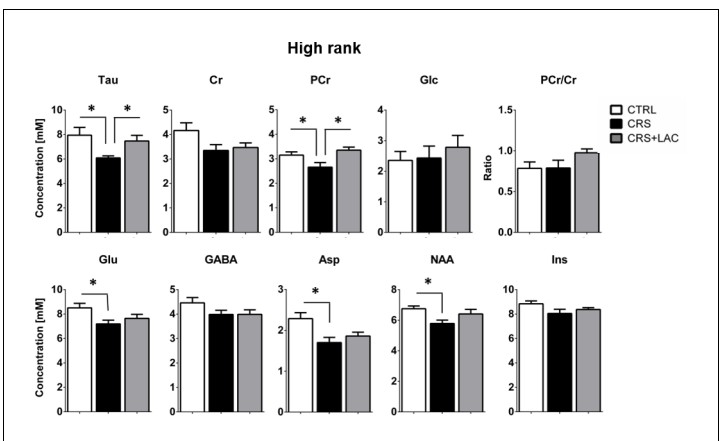

**Figure 5.** Effect of CRS and LAC treatment in CRS-treated mice on the accumbal neurochemical profile of high rank mice for metabolites with strong loading on Factor one.  Metabolites from Factor one with strong loading (above 0.5) include Tau, Cr, PCr, Glc, Glu, GABA, Asp, NAA and Ins. The ratio of PCr/Cr is shown as well. CRS induces a drop in Tau, Glu, PCr, Asp and NAA. Only CRS-induced reductions in Tau and PCr are restored by LAC treatment. The neurochemical profile obtained for low rank mice is reported in *Figure 5—figure supplement 1*. One-way ANOVA followed by LSD Fisher post-hoc test, *p<0.05, n = 5–6 per group.
The online version of this article includes the following figure supplement(s) for figure 5:

**Figure supplement 1.** Effect of LAC on the accumbal neurochemical profile of low rank mice after CRS.

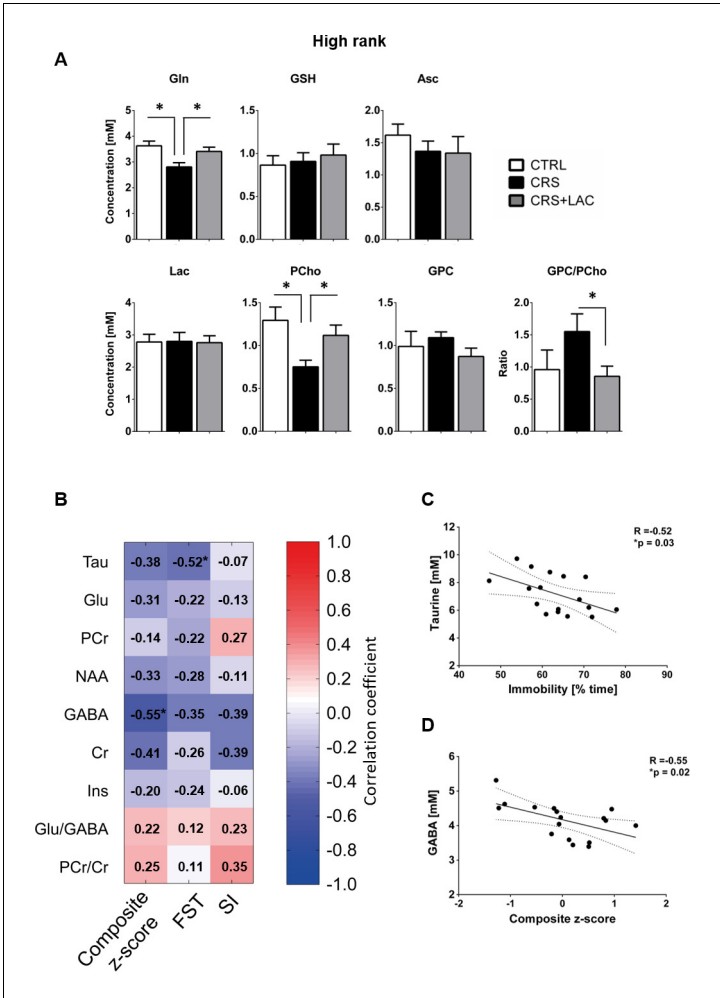

**Figure 6.** Effect of LAC on the accumbal neurochemical profile of high rank mice after CRS for remaining metabolites and associations between behavior and neurochemistry. (**A**) Metabolites with moderate loadings (0.4–0.5) from Factor one and remaining metabolites from Factors two and three included Gln, GSH, Asc, Lac, PCho and GPC. The ratio of GPC/PCho is also shown. CRS induces a drop in Gln and PCho, which are both restored after LAC treatment. The GPC/PCho ratio is also lowered after LAC administration. One-way ANOVA followed by LSD Fisher post-hoc test, *$p < 0.05$, n = 5–6 per group. (**B**) Correlation matrix between behavioral components and main metabolic targets of stress in the nucleus accumbens. Behavior included social interaction (SI) test, forced swim test (FST) and a composite behavior including both behaviors (Composite z-score). Each cell includes the Pearson's correlation coefficient with the associated color scaling. (**C**) Scatter plot of behavioral despair and accumbal taurine. (**D**) Scatter plot of depressive-like behavior and accumbal GABA. *$p < 0.05$, n = 16–18 per group.

least 5 weeks of cohabitation prior to carrying out the social confrontation tube test (see also *Larrieu et al., 2017*). This characterization may determine a different phenotype than studies in which social hierarchy is established within the first 2–3 weeks of cohabitation. Indeed, a recent study showed that during early cohabitation, dominance ranks of mice changed with repeated measurement, but became more stable between the 2nd and 3rd week of testing (*Varholick et al., 2018*). An additional issue to consider is that, for our analyses, we have grouped ranks 1 and 2 as high rank mice and ranks 3 and 4 as low rank. Although this grouping allows revealing statistical differences in the evaluated variables, each of the ranks in a tetrad home cage hierarchy may in fact lead to idiosyncratic phenotypes. In the future, it would be important to address vulnerability to stress for each specific rank in the colony.

LAC supplementation during the last week of the 3 week CRS protocol was efficient to protect high rank/vulnerable mice from the development of enhanced passive coping responses (i.e., higher

floating levels) in the FST. Importantly, LAC levels are markedly reduced in patients with major depressive disorder, particularly in those with treatment-resistant depression and higher reported rates of early life stress in the form of childhood trauma (*Nasca et al., 2018*). Our results in the FST are in line with previous rodent studies showing the ability of LAC to reduce immobility time in this test (*Wang et al., 2015*; *Bigio et al., 2016*; *Lau et al., 2017*; *Pulvirenti et al., 1990*) with several small clinical trials in humans reporting effectiveness of LAC treatment in amelioration of depressive symptoms (*Martinotti et al., 2011*; *Pettegrew et al., 2002*; *Zanardi and Smeraldi, 2006*; *Wang et al., 2014*; *Pettegrew et al., 2000*). Growing evidence showed the ability of LAC to ameliorate both social interaction and social avoidance at the chronic restraint stress and social defeat stress paradigms in mice with baseline anxiety-like behavior, increased systemic inflammation and decreased hippocampal volume (*Lau et al., 2017*; *Nasca et al., 2019*). Thus, the lack of LAC efficiency in reversing stress-induced deficits in social avoidance in mice with different social hierarchy range in the current study pave the way for future research in understading whether the social rank based on the SCTT relate to a mouse' behavior in a light-dark test that probably gets at the same anxiety versus resilient traits. It will be important to study whether social dominance and light dark traits co-occur in the same mice or whether these traits define distinct susceptible phenotypes that may show a differential responssivness to treatments. Moreover, the succesful LAC-induced reversal of increased floating in the FST would fit well with a view that the individual's NAc metabolic *millieu* is critically relevant for the animal's engagement in energetically-costly behaviors (*van der Kooij et al., 2018b*). Accordingly, depressive-like behaviors which are more fundamentally dependent on resources mobilization would be more likely to be reversed by LAC's energy support. Restoring normal metabolic function is, thus, more likely to affect behavior in the context set by the FST – that is fighting against an inescapable situation. This view fits well with the particular focus of this study on energy metabolism in the NAc. This postulate is supported by previous studies that showed that pharmacological manipulations leading to either impaired or boosted mitochondrial function in the nucleus accumbens were found to boost energetically-costly high rank behaviors during a social competition test between two male rats (*Hollis et al., 2015*; *van der Kooij et al., 2018a*).

Active coping in the FST depends upon dopamine actions in the NAc (*Tye et al., 2013*; *de Kloet and Molendijk, 2016*) and LAC has been shown to increase DA release in vivo (*Harsing et al., 1992*; *Tolu et al., 2002*) and to prevent a chronic stress-induced decrease in DA output in the NAc shell (*Masi et al., 2003*). In addition, stress has been shown to induce alterations in the accumbal oxidative stress system (*Della et al., 2012*; *Ignácio et al., 2017*), and LAC treatment to exert neuroprotective effects by inhibition of glial activation and oxidative stress in the striatum in an animal model of DAergic neuron damage (*Singh et al., 2016*). Furthermore, lipid peroxidation has been shown to be increased by immobilization stress in the rat striatum and L-carnitine to reduce associated striatal lipid peroxidation (*Méndez-Cuesta et al., 2011*). Interestingly, both in rodents (*Méndez-Cuesta et al., 2011*) and in zebrafish (*Marcon et al., 2019*), LAC was effective to reverse lipid peroxidation damage in stressed animals while being devoid of effect in controls.

Our in vivo $^1$H-MRS at 14 T identified key metabolites implicated in the response to chronic restrain stress and LAC treatment in the NAc. Among the nine metabolites loading highly in FA Factor one, five (i.e., taurine, phosphocreatine, glutamate, aspartate and NAA) were reduced by stress in high rank/vulnerable mice and two of them (i.e., taurine and phosphocreatine) reversed by LAC treatment. Taurine is a sulfur containing amino acid, with no involvement in protein synthesis, but with several functions ranging from antioxidant, signaling molecule and osmolyte (*Hansen et al., 2006*; *Yang et al., 2013*; *Wang et al., 2016*; *Jamshidzadeh et al., 2017*). Brain taurine concentrations have been shown to be reduced by chronic stress (*Barbosa Neto et al., 2012*) and hyperglycemic conditions (*Malone et al., 2008*). Although in our study we did not find stress-related changes in animals' glycemic state following CRS, the repeated stress schedule implemented in our CRS protocol is known to lead to increased blood glucose levels with every daily stress manipulation (e.g., *van der Kooij et al., 2018b*). It will be important to address whether this is the mechanism that leads to the reduction in accumbal taurine levels observed in our study, as well as exploring how metabolite changes induced by stress and LAC relate to mitochondrial function. Strikingly, in an early study, *Sershen et al. (1991)* specifically observed a reversal by LAC of ageing-induced reductions in taurine in the striatum, not of any other amino acid in the striatum or of taurine or other amino acids in any other brain region. Importantly, in our study, we also found that NAc taurine levels were the only ones from the measured metabolites that negatively correlated with passive coping

responses in the FST, reinforcing the link between levels of this amino acid and energetically-costly coping responses to adversity. In a recent 7T $^1$H-MRS in humans, we have recently found a negative correlation between trait anxiety and NAc taurine content (*Strasser et al., 2019*). Given the high link between trait anxiety and vulnerability to depression (*Sandi and Richter-Levin, 2009*; *Weger and Sandi, 2018*), our results support the interest in investigating the causal link between NAc taurine and its potential antidepressant actions.

Our finding that LAC increased phosphocreatine levels in high rank/vulnerable stressed mice is consistent with former reports indicating that LAC treatment increases phosphocreatine in the brain (*Castro et al., 2012*; *Smeland et al., 2012*; *Aureli et al., 1994*; *Aureli et al., 1990*; *Hansen et al., 2006*). Accordingly, LAC treatment likely improves the capacity of the brain to produce high-energy phosphates, which may be highly beneficial under conditions of disturbed energy metabolism. Several mechanisms have been implicated in the energy-boosting effects of LAC, many of them relating to an increased oxidative capacity of mitochondria through the direct release of oxidable fuel from LAC itself, or indirectly, in avoiding substrate inhibition of pyruvate dehydrogenase (PDH) by excess of AcCoA (*Smeland et al., 2012*; *Broderick et al., 1992*; *Panchal et al., 2015*; *Virmani et al., 1995*). Our results are in line with the idea of a restored mitochondrial function and support by LAC, visible through the increase in PCr, as well as taurine.

In addition, we observed that LAC restored levels of PCho and the ratio of GPC/PCho, which are disrupted by stress as well in high rank mice. PCho serves as a precursor of phosphatidylcholine (PtdCho), one of the main brain phospholipids, while GPC is its degradation product (*Morash et al., 1988*). The GPC/PCho ratio is thus considered to reflect the membrane turnover, typically increased in the case of neurodegeneration or excitotoxicity (*Nitsch et al., 1992*; *Kristián and Siesjö, 1998*). Increase in GPC can only arise from increased phospholipase activity, which is frequent during excitotoxic events and has been proposed to be a consequence of astrocyte activation (*Klein, 2000*; *Ha et al., 2014*). LAC and its deacetylated form L-carnitine (LC) are endogenous metabolites involved mainly in the transport and beta-oxidation of lipids. Exchange of LC with LAC and other acylcarnitines trough carnitine-acylcarnitine translocase (CACT) allows a bidirectional flow from cytoplasm into the inner mitochondrial matrix membrane for lipid oxidation. LAC supplementation could thus have a positive effect on phospholipid metabolism, by restoring normal balance between lipid degradation and synthesis.

A role for astrocytes in the LAC mechanism of action is also suggested by the normalization of stress-induced decreases in Gln content observed in the LAC-treated high rank group. Gln is mostly abundant in astrocytes (typically 80% of total concentration) due to their specific expression of glutamine synthetase (GS), which plays a key role in glutamate recycling at the synapse (*Norenberg and Martinez-Hernandez, 1979*; *Bak et al., 2006*). Astrocytic function is fundamental in the resilience to stress and regulation of extrasynaptic glutamate homeostasis (*Pellerin and Magistretti, 1994*; *Nasca et al., 2017*). For instance, LAC has shown effective control of astroglial cystine-glutamate exchanger (xCT), which is thought to improve mGlu2 function in hippocampus as a response to stress. A similar transcriptional response could be expected for glial GS, given its established responsiveness to glucocorticoids in stress (*Rozovsky et al., 1995*; *Carter et al., 2013*), an effect that could underlie the observed Gln changes. Even though the acetyl moiety of LAC has been shown to be utilized for the build-up of metabolic neurotransmitters synthesized from the TCA cycle (*Kuratsune et al., 2002*), LAC treatment in our study did not restore the stress-induced decrease of Glu, NAA and Asp observed in the nucleus accumbens of the high rank mice. This would suggest that LAC effects on these metabolites within the NAc may be secondary and part of an indirect and slower process. Nevertheless, as NAA, Glu and Asp measured with MRS reflect mainly neuronal metabolism (*Van den Berg et al., 1969*; *Bhakoo, 2012*), we can hypothesize that astrocytes are the first beneficiary of LAC supplementation. Astrocytes are indeed specifically shaped to uptake blood fatty acids, ketone bodies and acetate, and presumably LAC (*Blázquez et al., 1998*; *Valdebenito et al., 2016*; *Rae et al., 2012*).

Altogether, our findings highlight an accumbal metabolic signature for vulnerability to stress and response treatment. By implying an accumbal energy- and membrane metabolism process underlying the behavioral outcome, our study identifies molecular candidates responding in opposite direction to chronic stress and LAC treatment, opening possible mechanistic pathways underlying the anti-depressant-like effect of LAC. In particular, we underscore a strong association between NAc taurine and coping behaviors in an energetically-costly adversity task as well as antidepressant LAC

actions. However, it is important to note that our study of LAC effectiveness circumscribed to high rank mice. In the future, it will be important to establish whether LAC treatment would be effective to overcome emotional changes induced by a chronic stress regime capable of affecting low rank individuals and whether LAC treatment could be more effective in the high rank mice because of differential endogenous levels of LAC based on social hierarchy.

# Materials and methods

## Key resources table

| Reagent type (species) or resource | Designation | Source or reference | Identifiers | Additional information |
|---|---|---|---|---|
| Strain, strain background (*Mus musculus*) | Mouse: C57BL/6J | Charles River Laboratories | Crl:C57BL6/J | Male |
| Chemical compound, drug | Acetyl-L-carnitine | Sigma Aldrich | CAS Number:5080-50-2 | |
| Software, algorithm | Matlab v.9.6 | The MathWorks | RRID:SCR_001622 | |
| Software, algorithm | Observer 11.0 | Noldus, Information Technology | RRID:SCR_004074 | |
| Software, algorithm | Ethovision 11.0 XT | Noldus, Information Technology | RRID:SCR_000441 | |
| Software, algorithm | Prism 6 | GrahpPad | RRID:SCR_002798 | |
| Software, algorithm | LCModel | LCModel | RRID:SCR_014455 | |
| Software, algorithm | SPSS version 21 | IBM | https://www.ibm.com/analytics/fr/fr/technology/spss/ | |

## Animals

Six-week-old male C57BL/6J mice were purchased from Charles River Laboratories and, upon arrival, they were housed in groups of four per cage and allowed to acclimate to the animal facility for one week. Mice were weighed at arrival and monitored throughout the experiments. Cages consisted in standard Plexiglass filter-top cages in a temperature ($23 \pm 1°C$) and humidity (40%) controlled environment with normal 12 hr day-light cycle. Animals had ad libitum access to water and standard rodent chow diet. All experiments were performed with the approval of the Cantonal Veterinary Authorities (Vaud, Switzerland) and carried out in accordance with the European Communities Council Directive of 24 November 1986 (86/609EEC).

## Experimental design

One week after arrival, mice were tested for their anxiety and locomotor behaviors in an EPM and OF (*Figure 1A*). After four weeks of cohabitation, a SCTT was used to reveal individual ranks within the home cage tetrad (*Larrieu et al., 2017*). Subsequently, one group of the animals was subjected to a CRS protocol for 21 days, while the remaining non-stressed animals were submitted to daily handling and body weighting. The impact of chronic stress on behavior was tested in the SI test (CRS day 20). The experiment performed to investigate the ability of LAC treatment on behavioral and metabolic outcomes of CRS, included an additional group treated with LAC from CRS day 15. In addition to the SI test, animals were tested in FST (CRS day 21). Subsequently, $^1$H-MRS was performed at the end of the protocol, 24 hr after the FST (day 22).

## Elevated plus maze test

Animals were placed into a maze made from black PVC with a white floor. The apparatus consisted of an elevated central platform ($5 \times 5$ cm$^2$) at 65 cm from the ground, from which four opposing arms extended. Two of the arms were open ($30 \times 5$ cm$^2$) and lit with 14–15 lx while the two others were closed ($30 \times 5 \times 14$ cm$^3$) with reduced light intensity 3–4 lx. Animals were introduced in the maze facing the wall at the end of closed arms and left freely moving for 5 min. The mice were video-recorded from above the arena and tracking analyses performed with the Ethovision 11.0 XT software (Noldus, Information Technology) to determine the time spent in open and closed arms.

## Open field test

The OF consisted of a rectangular arena (50 × 50 × 40 cm$^3$) illuminated with dimmed light (30 lx). Mice were introduced near the wall of the arena and allowed to explore for 10 min. Analyses were performed using a tracking software (Ethovision 11.0 XT, Noldus, Information Technology) by drawing a virtual zone (15 × 15 cm$^2$) in the center of the arena defined as the anxiogenic area. Several parameters were analyzed, including the total distance travelled and the time spent in the different zones.

## Social confrontation tube test

The SCTT test was performed as previously described (*Larrieu et al., 2017*; *Wang et al., 2011*). Mice were housed together in groups of four during 5 weeks prior to the test to allow enough time for the social hierarchy to be stabilized in the home cage. First, each mouse was independently habituated to cross over a plastic tube (Plexiglas of 3 × 30 cm$^2$, diameter x length) on several trials in two consecutive days. Then, evaluation of social rank took place during eight consecutive days. Specifically, two mice were smoothly guided by the tail to get into each side of the tube for a pairwise confrontation. Once the two mice reached the middle of the tube, the tail was released, and the time spent was recorded until one the mice (the most subordinate) retracted out from the tube. The four mice from the same cage were opposed using a round-robin design that led to six face-to-face trials per day. The tube was cleaned with 70% ethanol and dried after each session. The designation of a winner after each of the six possible pairs of confrontations per cage allowed to rank each mouse by its winning times, varying between 0–3. This value was then divided by three and multiplied by 100 to directly obtain the winning percentage. This percentage was then used to determine the index of dominance ranging from 1 (highest winning percentage, that is highest rank) to 4 (lowest winning percentage, that is lowest rank) for each cage. In order to reflect the specificity of the observed dominant and subordinate behavior to the SCTT (*Varholick et al., 2018*), animals with highest index of dominance (ranks 1 and 2) are referred to as 'high rank' herein, whilst animals with lowest index of dominance (ranks 3 and 4) are referred to as 'low rank'.

## Chronic restraint stress

This protocol involved 21 days of chronic retrain stress (CRS) and was adapted from *Lau et al. (2017)*; *Nasca et al. (2015)*. Animals were introduced head first into a 50 ml Falcon tube (11.5 cm in length; diameter of 3 cm) in which the cap was removed. Each restrain tube contained 3 0.4 cm air holes to allow the air to reach the nose of the mouse. Paper was added at the other extremity to adjust the physical constraint to the mouse body size and allowing the tail to reach the open space. The mice were subjected to this restrained environment for two consecutive hours every day for a period of 21 days. Control mice were left undisturbed in their home cage except for handling and body weighting each day for 21 days.

## Social avoidance test (social interaction test)

Each animal was introduced into a 40 × 40 × 30 cm white arena containing an unfamiliar old breeder CD1 male mouse (social target) confined in a cylindrical drum with wire mesh placed near one of the arena walls. The test consists in two phases. First, the experimental mouse was allowed to freely explore for 2.5 min the arena when the social target was absent (the arena contained only the drum). Then, the target mouse was introduced in the drum for a 2.5 min interaction session. A social avoidance score was calculated as previously described in *Larrieu et al. (2017)*. The mice were video-recorded from above the arena and tracking analyses performed with the Ethovision 11.0 XT software (Noldus, Information Technology).

## Forced swim test

Each animal was introduced into a cylinder (15 cm diameter, 28 cm in height) filled with 5 L 25°C tap water. The level of water was sufficiently high to avoid any contact of the mouse with the bottom of the enclosure and low enough to avoid any possible escape. Animal's motion was tracked with a camera positioned on top of the setup and recorded for 6 min. Immobility time was quantified using the Observer XT software (Noldus, Information Technology).

## Composite behavior

Using MATLAB (Version 9.6, The MathsWorks Inc, Natick, MA), a behavioral composite z-score was calculated by averaging the z-score of the social avoidance score in the SI test and the z-score of the % time spent immobile in the FST. Both z-scores were calculated with the MATLAB function *normalize*, using the option argument *zscore*, which divides the difference between each sample and the sample average by the sample's standard deviation.

## Acetyl-L-carnitine treatment

LAC was purchased from (Sigma Aldrich). Mice received LAC in the drinking water at a concentration of 0.3%. The treatment started on day 15 of the 21 day-CRS protocol and continued till the end of the experiment. Control groups received regular tap water available ad libitum. In order to maximize the potential therapeutic effects of LAC on the stress-induced depressive-like behavior of mice (*Nasca et al., 2013*), a 7 day treatment was preferred over a previously reported 3 day protocol (*Lau et al., 2017*; *Nasca et al., 2017*). One bottle per cage of 4 mice was used during LAC treatment. Liquid consumption was monitored every day and analyzed following normalization according to the body weight of the four animals in the cage.

## $^1$H-magnetic resonance spectroscopy

In vivo spectroscopy experiments targeting the NAc were performed in anesthetized mice as previously described (*Larrieu et al., 2017*). Animals were monitored for body temperature (rectal probe and circulating water bath) and respiration (small animal monitor system: SA Instruments Inc, New York, NY, USA) under 1.3–1.5% isoflurane anesthesia mixed with 50% air and 50% $O_2$. Physiological parameters were maintained at $36.5 \pm 0.4°C$ and breathing rate ranged between 70–100 rpm. Animals were scanned in a horizontal 14.1T/26 cm Varian magnet (Agilent Inc, USA) with a homemade $^1$H surface coil. A set of Fast Spin Echo (FSE) images of the brain was acquired for localizing the Volume of Interest (VOI) of the $^1$H-MRS scan. Acquisition was done using the spin echo full intensity acquired localized (SPECIAL) sequence (*Mlynárik et al., 2006*) in the VOI of $1.4 \times 4.1 \times 1.2$ mm$^3$ (TE/TR = 2.8/4000 ms) including the bilateral NAc after field homogeneity adjustment with FAST (EST)MAP (*Gruetter and Tkác, 2000*). The obtained spectra ($20 \times 16$ averages) were then frequency corrected, summed and quantified using LCModel (*Provencher, 2001*). Full width at half maximum (FWHM) was used as the output of the LCModel. Concentrations were referenced to the water signal and fitting quality assessed using Cramer-Rao lower bounds errors (CRLB) (*Cavassila et al., 2001*). A CRLB value below 20% was used as cutoff for high concentration metabolites, while low concentration metabolites were not considered reliable above a CRLB of 50%.

## Statistics

All values are represented as mean ± SEM. Results from EPM and OF were analyzed using unpaired Student t-tests. Results from the SCTT were analyzed with one-way analysis of variance (ANOVA), with social rank as fixed factor, followed by a Bonferroni corrected *post hoc* test when appropriate. Behaviors in the SI test and FST were analyzed with a two-way ANOVA, using stress and social rank as fixed factors. In the LAC experiment, behavioral and spectroscopy results were analyzed using one-way ANOVA. Cumulative weight gain was analyzed using a repeated measure two-way ANOVA with time and group as fixed factors. Analyses were followed by Bonferroni post hoc correction when appropriate. Correlations analyses were performed using a Pearson's correlation coefficient. All statistical tests were performed with GraphPad Prism (GraphPad software, San Diego, CA, USA) using a critical probability of p<0.05. Statistical analyses performed for each experiment are summarized in each figure legend indicating the statistical test used, sample size ('n'), as well as degree of freedom, F and P values.

## Factor analysis

Factor analysis was used as previously described (*Larrieu et al., 2017*) using IBM SPSS Statistics version 21 to allow statistical tests using the metabolite's latent variables as dependent variables in NAc. A linear combination of the dependent variables is generated in order to reduce the noise caused by the high number of variables. Missing values were avoided by using mean value imputation before the computation of correlation matrices, to ensure positive definiteness. A total of three

factors was chosen for the NAc after analyzing the scree plots, using principal axis factoring. This resulted in a total of variance explained of 52% without rotation and omitting coefficients below 0.4.

## Acknowledgements

This project has been supported by grants from the Swiss National Science Foundation [31003A-152614 and −176206; NCCR Synapsy (51NF40-158776 and −185897)], the European Union's Seventh Framework Program for research, technological development and demonstration under grant agreement no. 603016 (MATRICS), the EPFL-Jebsen Research Program and intramural funding from the EPFL to CS. The funding sources had no additional role in study design, in the collection, analysis and interpretation of data, in the writing of the report or in the decision to submit the paper for publication. This paper reflects only the authors' views and the European Union is not liable for any use that may be made of the information contained therein.[1]H-MRS experiments were also supported financially by the Center for Biomedical Imaging (CIBM) of the University of Lausanne (UNIL), University of Geneva (UNIGE), Geneva University Hospital (HUG), Lausanne University Hospital (CHUV), Swiss Federal Institute of Technology (EPFL) and the Leenaards and Louis-Jeantet Foundations.

## Additional information

### Funding

| Funder | Grant reference number | Author |
| --- | --- | --- |
| Swiss National Science Foundation | 31003A-152614 | Carmen Sandi |
| Swiss National Science Foundation | 31003A-176206 | Carmen Sandi |
| Swiss National Science Foundation | NCCR Synapsy 51NF40-158776 | Carmen Sandi |
| Swiss National Science Foundation | NCCR Synapsy 51NF40-185897 | Carmen Sandi |
| EU Seventh Framework Programme | 603016 | Carmen Sandi |
| École Polytechnique Fédérale de Lausanne | Jebsen Research Program | Carmen Sandi |
| Center for Biomedical Imaging | | Rolf Gruetter |

The funders had no role in study design, data collection and interpretation, or the decision to submit the work for publication.

### Author contributions

Antoine Cherix, Conceptualization, Formal analysis, Investigation, Visualization; Thomas Larrieu, Conceptualization, Formal analysis, Supervision, Investigation; Jocelyn Grosse, João Rodrigues, Conceptualization, Formal analysis, Investigation; Bruce McEwen, Carla Nasca, Conceptualization, Supervision; Rolf Gruetter, Conceptualization, Supervision, Funding acquisition; Carmen Sandi, Conceptualization, Supervision, Funding acquisition, Project administration

### Author ORCIDs

Antoine Cherix ![ORCID] https://orcid.org/0000-0002-4168-8273
Carmen Sandi ![ORCID] https://orcid.org/0000-0001-7713-8321

### Ethics

Animal experimentation: All experiments were performed with the approval of the Cantonal Veterinary Authorities (Vaud, Switzerland) and carried out in accordance with the European Communities Council Directive of 24 November 1986 (86/609EEC).

Decision letter and Author response
Decision letter https://doi.org/10.7554/eLife.50631.sa1
Author response https://doi.org/10.7554/eLife.50631.sa2

## Additional files

### Supplementary files
• Transparent reporting form

### Data availability
All data generated or analysed during this study are included in the manuscript and supporting files.

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
