## [Decision Letter]

**Acceptance summary:**

Depression is a common mental disorder that is a major cause of disability worldwide. This emphasizes the need to get a better understanding of the underlying mechanisms. While prolonged stress-exposure is an important risk factor for depression, not all individuals are vulnerable to develop depression after stress-exposure. Here, the authors tested in mice the relationship between social rank and the vulnerability to develop depressive like behaviour after exposure to chronic stress. The authors report in mice that high rank animals are more vulnerable to chronic stress-exposure in comparison to low rank animals. In these more sensitive mice, the levels of several energy-related metabolites in the nucleus accumbens were reduced. Interestingly, Acetyl L-carnitine – a mitochondrial-boosting supplement – administered via the drinking water was able to partially reduce behavioural effects and brain metabolic alterations after chronic stress and in high rank mice. This work illustrates a role for energy metabolism in the vulnerability for developing chronic-stress evoked depressive like behaviour.

**Decision letter after peer review:**

Thank you for submitting your article "Metabolic signature in nucleus accumbens for anti-depressant-like effects of acetyl-L-carnitine" for consideration by *eLife*. Your article has been reviewed by three peer reviewers, and the evaluation has been overseen by a Reviewing Editor and Christian Büchel as the Senior Editor.

The reviewers have discussed their reviews with one another and the Reviewing Editor has drafted this decision to help you prepare a revised submission.

In this manuscript, the authors report that dominant, but not subordinate, mice exhibit a vulnerable phenotype after chronic stress exposure. Dominant animals also expressed reduced levels of energy-related metabolites in the nucleus accumbens. Alterations in metabolic signature and behaviour were counteracted by treatment with Acetyl-L-Carnitine. While the reviewers generally find the study interesting, well-conducted and find the manuscript well written, they also raise a number of issues.

In particular, the reviewers ask you to consider statistics, details on experimental procedures and potential motor effects of Acteyl-L-Carnitine (Reviewer 1); Discuss and consider the definition of dominant/subordinate (Reviewer 2) and the forced swim test (Reviewer 3). Additional data might be helpful to address some of the reviewers’ issues.

Please address the issues raised by the reviewers in a point-by-point reply.

Reviewer #1:

In this paper, the authors test the hypothesis that chronic stress vulnerability and consequential development of depressive-like symptoms is linked to energy metabolic changes in the nucleus accumbens, which can be reversed by LAC supplementation. The authors based their hypothesis on previous results involving the NAc in a protocol to induce depressive-like symptoms based on chronic social defeat and preclinical studies using LAC to alleviate depressive-like symptoms. To test their hypothesis, they used a variant of the chronic stress protocol using restrain stress for 21 days, and administered LAC via water supplementation the last 7 days of the protocol. To assess the impact of both the stress and the treatment in depressive-like symptoms, they used a social avoidance and a forced swimming test, and they analyzed up to 20 metabolites using in vivo H-MRS related to mitochondrial energy metabolism, neurotransmission and lipid peroxidation.

The study is interesting, well conducted and well written. There are however a few considerations to be taken into account to improve the quality of the manuscript and the completeness of the data to support the conclusions.

1) Effects of LAC treatment on depressive-like behaviors in dominant and subordinate mice. Although the sample size of the data is small (n = 6-7), the experiments were able to replicate the predictability of the social status based on the anxiety phenotype and the vulnerability of the dominant mice to chronic-stress induced depressive-like deficits like social avoidance (Figure 1). Interestingly, they failed to see the basal difference in latency to immobility in the forced swimming test between dominant and subordinate male mice reported by previous published data (Horii et al., 2017). However, the trend between the control, CRS and CRS+LAC conditions is the same between dominant and subordinate males, as the reduction in the% of immobility appears to be the same between the CRS and the CRS+LAC dominant and subordinate groups. If the authors wish to report conclusions of their effect of the LAC treatment across dominant and subordinate mice (as they do in their Results and Discussion sections), they should carry on a two-way ANOVA with post-hoc analysis with social status and group condition as the variables in all of their behavioral tests (social avoidance, FST and composite behavior) (Figure 2).

2) Moreover, seeing that LAC supplementation appears to decrease% cumulative weight gain and% of immobility of the stressed mice, one could argue that the effect of LAC could be due to an increase in locomotor activity of the mice which, on one hand could be considered a treatment for depressed locomotor activity but on the other hand could also limit the therapeutic potential of the mitochondrial boost supplementation. While Lau and colleagues (2017) didn't see effects of LAC in immobility in non-stressed mice and no effect should be expected in the experiments presented in this article, I would suggest the authors to provide a graph with the total distance travelled in the FST as a supplementary figure to discard the motor effects of LAC or an extra open field test post treatment.

3) In terms of the LAC dose administration, the authors report to have used drinking water supplementation with a concentration of 0.3%. It is not clear, however, if at the time of administration, the animals were singled or grouped housed (in groups of 4 as during the cohabitation). If possible, a water intake report should be presented in order to report a clear dosage of the supplement per animal.

4) Changes in NAc metabolites in response to chronic stress and LAC supplementation

The use of in vivo H-MRS allows to explore multiple metabolites and neurochemical compounds at the same time in dominant and subordinate males, which englobe processes known to be affected by LAC like mitochondrial beta-oxidation, protein acetylation, lipid peroxidation and neurotransmitter composition. According to their results, chronic stress in dominant mice affects different metabolites related to mitochondrial transport and glutamatergic neurotransmission.

Interestingly, they saw a decrease in taurine levels in the CRS dominant group that was restored to control levels by LAC supplementation (Figure 5). While the authors discuss the negative correlation between NAc taurine levels and traits like anxiety and immobility, they don't discuss the surprising nature of their results. Taurine is a mitochondrial transport protein that is implicated in oxidative stress, and typically increased in situations that require high mitochondrial activity like chronic restrain stress and decreased by hyperglycemia. However, in these results, CRS decreases taurine levels without changes in glucose levels. Furthermore, this only is seen in dominant males. A deeper discussion of these results is recommended to accentuate the importance of the findings regarding the changes in taurine.

5) Regarding the main effects of LAC in correcting the defects in mitochondrial-related metabolites, an extra experiment measuring complex I-II activity in the NAc (following the experiments by the corresponding author in Hollis et al., 2015) would be appropriate to support the conclusions.

Reviewer #2:

Cherix and coworkers investigated the impact of acetyl-L-carnitine treatment on stress induced depression-like behavior in mice characterized according to the rank assessed in a tube test. The experiments are conducted to a high technical standard and the manuscript is very well written. Although the overall findings are interesting and novel (and this study would be of interest for the audience of *eLife*), there is one major and other specific concerns that should be addressed:

1) Even after reading several times this manuscript (and the previous Current Biology 2017 paper), I still struggle with the nomenclature used, i.e. dominant and subordinate, and how the individual rank is assessed. Below several concerns on this issue, and an overall recommendation to use a different nomenclature and avoid interpreting the data in favor of social status differences.

– I’m still unclear of what the Figure 1C is actually measuring, and how the index of dominance (Figure 1B) was calculated. The Materials and methods states "the time spent was recorded until one of the mouse (the most subordinate) retracted out from the tube". This definition assumes a priori that a mouse rapidly retracting from the tube is subordinate, and does not take into account what the other mouse in the test tube is actually doing (isn`t the purpose of this test to evaluate concomitantly the behavior of 2 mice to assess their relative rank?). I provide below several alternative (speculative) interpretations: i) high time spent in the tube implies extended social affiliation between 2 mice, and not social confrontation; ii) low time implies high fear of the tube per se, independently from the partner presence; iii) high time implies enhanced fear response manifested as freezing/immobility, again independently from the social interaction.

– The conclusion that the rank in the tube test is equivalent to social rank has not been confirmed with behavioral observations of agonistic interaction in the home cage. Mice were housed for 5 weeks prior to the tube test to "allow enough time for the social hierarchy to be stabilized". However, there is no record of the agonistic interactions and the social rank in the home cage. How do they know that the hierarchy in the home cage is reflected in the rank in the tube test? The experience with B6 is that very often the hierarchy is not as linear as the test tube rank will suggest.

– The way "dominant" and "subordinate" are defined trough the manuscript is methodologically concerning (comparing ranks 1+2 to ranks 3+4 is equivalent to a median split of the population; a questionable analytical approach) and biologically unclear (what`s the relevance of grouping 1/2 and 3/4 for the sociobiology of mice, which are known to be highly despotic? Classic paper https://academic.oup.com/jmammal/article/36/2/299/960816).

– Authors should also consider and discuss the results of this study (https://www.ncbi.nlm.nih.gov/pmc/articles/PMC5920077/) that used a similar tube test protocol (as well as their review of the relevant literature).

– Overall, I strongly suggest to avoid the definition of "dominant" and "subordinate" in the manuscript and to analyze their very nice data according to a more neutral "high rank" and "low rank" in the tube test (or even better to compare the 4 ranks individually, e.g. alpha, beta, etc) and try to determine, or at least speculate, of what that rank really means. As they and others have demonstrated, there is some very interesting biology associated with the rank assessed in this test. It clearly reflects an individual trait (similar to trait anxiety measured in the EPF for example?) that may or may not be related to social status, or social rank. Indeed, there is a vast literature showing that when social rank is assessed using behavioral observations of social interaction, subordinate mice manifested higher anxiety and depression-like behavior when compared to dominant mice, a finding which is opposite when ranks is assessed in the tube test (Figure 1).

2) Figure 2. LAC treatment significantly improved stress-induced immobility (albeit the% change is modest) but had no effect in the social interaction test. Therefore, authors should be very careful in using strong statements on the "anti-depressant" effect of this drug.

3) I see little added value in the use of the composite behavioral score of just 2 tests, out of which only one show a significant difference between groups. Essentially the score will be driven by the FST. Unless authors will be able to provide a strong rationale for its significance, this score should be removed.

4) Figure 1F; please include actual total time in the interaction zone and the corners.

5) Figure 1E; please include time in the center, time in the outer ring or corners etc. The significance of latency to enter the center as the most important measure of anxiety is questionable.

Reviewer #3:

The authors examined the potential antidepressant effects of Acetyl-L-Carnitine in dominant and subordinate mice exposed to chronic restraint stress. Dominant and subordinate mice were classified based on their hierarchical status. Following chronic stress, dominant mice show vulnerability in social interaction and in the forced swim test. Treatment with carnitine reversed the deficits seen in the Porsolt swim test but not social avoidance. The authors then used MRS and identified in the NAc some metabolites that are decreased by stress and rescued by carnitine. Some of these metabolites are related to mitochondrial function. Despite being correlational, this discovery approach has definitely some merits as it has identified some targets in the NAc that could be followed on with functional studies, in the hope that they play a major role in depression. While we liked this work, we still have a few major concerns that the authors need to address:

1) The authors are relying too much on the forced swim test to demonstrate that L-Carnitine is an antidepressant drug. As the authors likely know, this behavioral test has received a lot of criticism lately and is definitely not a test for depression. Adding more behavioral tests -may be tests that require energy and coping- is necessary to strengthen this manuscript.

2) The EPM data shown in Figure 1: Animals in this test spend 4-8% of their time in the open arms. This is an extremely low time that will not allow the experimenters to appreciate the effects of anxiogenic treatments.

3) OF data shown in Figure 1: why only latency to enter the center is reported? Usually, it is the time spent in the center of the OF that is reported

4) Why rank 2 was considered dominant? If that is the case, rank 3 is also dominant over rank 4, right? Did the authors think about examining just the extremes (rank 1 vs. rank4)?

[Editors' note: further revisions were suggested prior to acceptance, as described below.]

Thank you for resubmitting your work entitled "Metabolic signature in ccumbens for anti-depressant-like effects of acetyl-L-carnitine" for further consideration by *eLife*. Your revised article has been evaluated by Christian Büchel (Senior Editor) and a Reviewing Editor.

The manuscript has been improved but there are some remaining issues that need to be addressed before acceptance, as outlined below.

1) While the authors changed the nomenclature from dom/sub to high/low rank in the Result section/figures, the interpretation of what that rank reflects is unchanged in the Abstract, Introduction and large part of the Discussion. As currently written, the role of status/rank in the observed phenotype is confusing. If authors consider social rank a trait (genetically encoded?) rather than a state, they should be consistent through the manuscript.

2) Authors did not address a main concern: high and low rank groups simply reflect a median split of the population. A visual inspection of Figure 1 clarifies that ranks 2 and 3 are very similar, essentially indistinguishable from each other. Because the N is low, they could not analyze each rank separately, and rank 1 vs rank 4 are not statistically different (Author response images).

3) The authors did not provide any justification for the use of the composite behavioral score (except for convenience of having a second measure of "global" depression-like behavior).

4) Finally, the evidence in favor of high rank mice being characterized by high anxiety is limited to only 1 behavioral test, the EPM.

I would like to ask the authors to address these comments by thoroughly discussing the strengths/limits of the ranking dominance classifications.

---

## [Author Response]

Reviewer #1:[…] 1) Effects of LAC treatment on depressive-like behaviors in dominant and subordinate mice. Although the sample size of the data is small (n = 6-7), the experiments were able to replicate the predictability of the social status based on the anxiety phenotype and the vulnerability of the dominant mice to chronic-stress induced depressive-like deficits like social avoidance (Figure 1). Interestingly, they failed to see the basal difference in latency to immobility in the forced swimming test between dominant and subordinate male mice reported by previous published data (Horii et al., 2017). However, the trend between the control, CRS and CRS+LAC conditions is the same between dominant and subordinate males, as the reduction in the% of immobility appears to be the same between the CRS and the CRS+LAC dominant and subordinate groups. If the authors wish to report conclusions of their effect of the LAC treatment across dominant and subordinate mice (as they do in their results and Discussion sections), they should carry on a two-way ANOVA with post-hoc analysis with social status and group condition as the variables in all of their behavioral tests (social avoidance, FST and composite behavior) (Figure 2).

We convey with the reviewer that in order to claim a differential effect between high rank and low rank mice, regarding the efficacy of the LAC treatment, the observation of an interaction in the behavioral tests is required. As the interaction effect is not significant, we have followed our established goal for the study which was/is to investigate whether LAC treatment may prevent the emerging of depression-like behaviors and metabolic dysregulation in the NAc induced by CRS.

Accordingly, we have made this aim more explicit in the text and specified in the Results section as well that our focus in the LAC experiment was to examine results in high rank mice. All the results from low rank animals in the LAC experiment have been moved to the figure supplements (i.e., *eLife* ‘child’ figures; Figure 5—figure supplement 1). All previous reference to the comparison of LAC treatment effects in high vs low rank animals have now been deleted from the text.

2) Moreover, seeing that LAC supplementation appears to decrease% cumulative weight gain and% of immobility of the stressed mice, one could argue that the effect of LAC could be due to an increase in locomotor activity of the mice which, on one hand could be considered a treatment for depressed locomotor activity but on the other hand could also limit the therapeutic potential of the mitochondrial boost supplementation. While Lau and colleagues (2017) didn't see effects of LAC in immobility in non-stressed mice and no effect should be expected in the experiments presented in this article, I would suggest the authors to provide a graph with the total distance travelled in the FST as a supplementary figure to discard the motor effects of LAC or an extra open field test post treatment.

In order to address the reviewer’s concern and to evaluate potential effects of the treatments on general animal locomotion, we have measured the distance travelled by mice in the Social Interaction test (measurement in the FST gives similar -though inverse- results as the ones provided in Figure 2E). Locomotor activity was increased by CRS and CRS+LAC did not significantly differ in locomotion from CRS ones, though they tended to move less when the target animal was not present (Figure 2G). If anything, LAC-treated mice tended to move slightly less than CRS and non-LAC treated mice, supporting the idea that the reduced floating induced by LAC treatment in CRS-treated mice in the FST is not explained by a general increase in locomotor activity induced by the treatment.

3) In terms of the LAC dose administration, the authors report to have used drinking water supplementation with a concentration of 0.3%. It is not clear, however, if at the time of administration, the animals were singled or grouped housed (in groups of 4 as during the cohabitation). If possible, a water intake report should be presented in order to report a clear dosage of the supplement per animal.

We have now clarified this point in the Materials and methods section, as follows:

“One bottle per cage of 4 mice was used during LAC treatment. Liquid consumption was monitored every day and analyzed following normalization according to the body weight of the 4 animals in the cage.”

The water intake report has been described in the Result section, as follows:

“Animals exposed to CRS displayed a significant decrease in cumulative body weight gain, regardless of their social rank, that was not counteracted by LAC supplementation (Figure 2B; in high rank: Stress effect F_2,10_=45.0, p<.0001 and Figure 2—figure supplement 1A; in low rank: Stress effect F_2,10_= 20.4, p<.001). Importantly, whereas stress led to an increase in liquid consumption (+15 ± 12%) in the stressed groups (Figure 2C and Figure 2—figure supplement 1B), there was no difference in liquid consumption between the CRS and CRS+LAC groups (Figure 2C)”.

With statistics described in the figure legend, as follows:

“Figure 2 (C) Daily water intake during the LAC treatment period (given during the last week of the CRS protocol) normalized by total body weight of the 4 mice per cage (Group effect: F2,4=17.0, *P<.05; Interaction: F14,28=0.90, P>.05; time effect: F7,14=1.24, P>.05, repeated measures two-way ANOVA; n=3 cages per group). Thus, water intake data represents the cage average value.”

4) Changes in NAc metabolites in response to chronic stress and LAC supplementationThe use of in vivo H-MRS allows to explore multiple metabolites and neurochemical compounds at the same time in dominant and subordinate males, which englobe processes known to be affected by LAC like mitochondrial betation, protein acetylation, lipid peroxidation and neurotransmitter composition. According to their results, chronic stress in dominant mice affects different metabolites related to mitochondrial transport and glutamatergic neurotransmission.Interestingly, they saw a decrease in taurine levels in the CRS dominant group that was restored to control levels by LAC supplementation (Figure 5). While the authors discuss the negative correlation between NAc taurine levels and traits like anxiety and immobility, they don't discuss the surprising nature of their results. Taurine is a mitochondrial transport protein that is implicated in oxidative stress, and typically increased in situations that require high mitochondrial activity like chronic restrain stress and decreased by hyperglycemia. However, in these results, CRS decreases taurine levels without changes in glucose levels. Furthermore, this only is seen in dominant males. A deeper discussion of these results is recommended to accentuate the importance of the findings regarding the changes in taurine.

We thank the reviewer for bringing up this mechanistic perspective. Indeed, stress-induced hyperglycemia during daily stress exposure may have contributed to the observed changes in NAc taurine levels. Accordingly, the lower levels of NAc taurine found in stressed high rank mice have been now further addressed with respect to hyperglycemia, by including the following paragraph in the Discussion section:

“Taurine is a sulfur containing amino acid, with no involvement in protein synthesis, but with several functions ranging from antioxidant, signaling molecule and osmolyte (Hansen et al., 2006; Yang et al., 2013; Wang et al., 2016; Jamshidzadeh et al., 2017). Brain taurine concentrations have been shown to be reduced by chronic stress (Barbosa Neto et al., 2012) and hyperglycemic conditions (Malone et al., 2008). […] It will be important to address whether this is the mechanism that leads to the reduction in accumbal taurine levels observed in our study.”

As to the effects of chronic stress in mitochondrial function, given that there are brain region specific differences typically found in the literature, we would prefer not to include this aspect, until its careful consideration in future studies, for clarity of the message and discussion.

5) Regarding the main effects of LAC in correcting the defects in mitochondrial-related metabolites, an extra experiment measuring complex I-II activity in the NAc (following the experiments by the corresponding author in Hollis et al., 2015) would be appropriate to support the conclusions.

Although the measurement of mitochondrial respiration for complexes I and II as in Hollis et al., 2015, could provide further insights on whether mitochondrial function is specifically affected by the treatment, and we agree of interest on its own, we would like to address these and several other potential mechanisms related to mitochondria and ER, and their interactions in the context of stress and nutritional supplements, such as LAC, in future studies that go in-depth into mechanistic understanding. At this point, and given the lengthy investment that such an experiment would require, we have opted by focusing on the metabolic characterization of metabolites that can be quantified with ^1^H-MRS, in order to progress on potential endpoints and biomarkers that can be followed up as well in human studies. Therefore, and hoping that it is ok with the reviewers and editors, and due to the actual lack of time to perform such lengthy experiment for the requested revision, we have addressed now this point by including the following comment in the Discussion section:

“It will be important to address whether this is the mechanism that leads to the reduction in accumbal taurine levels observed in our study, as well as exploring how metabolite changes induced by stress and LAC relate to mitochondrial function.”

Reviewer #2:[…] 1) Even after reading several times this manuscript (and the previous Current Biology 2017 paper), I still struggle with the nomenclature used, i.e. dominant and subordinate, and how the individual rank is assessed. Below several concerns on this issue, and an overall recommendation to use a different nomenclature and avoid interpreting the data in favor of social status differences.– I’m still unclear of what the Figure 1C is actually measuring, and how the index of dominance (Figure 1B) was calculated. […] I provide below several alternative (speculative) interpretations: i) high time spent in the tube implies extended social affiliation between 2 mice, and not social confrontation; ii) low time implies high fear of the tube per se, independently from the partner presence; iii) high time implies enhanced fear response manifested as freezing/immobility, again independently from the social interaction.– The conclusion that the rank in the tube test is equivalent to social rank has not been confirmed with behavioral observations of agonistic interaction in the home cage. Mice were housed for 5 weeks prior to the tube test to "allow enough time for the social hierarchy to be stabilized". However, there is no record of the agonistic interactions and the social rank in the home cage. How do they know that the hierarchy in the home cage is reflected in the rank in the tube test? The experience with B6 is that very often the hierarchy is not as linear as the test tube rank will suggest.– The way "dominant" and "subordinate" are defined trough the manuscript is methodologically concerning (comparing ranks 1+2 to ranks 3+4 is equivalent to a median split of the population; a questionable analytical approach) and biologically unclear (what`s the relevance of grouping 1/2 and 3/4 for the sociobiology of mice, which are known to be highly despotic? Classic paper https://academic.oup.com/jmammal/article/36/2/299/960816).– Authors should also consider and discuss the results of this study (https://www.ncbi.nlm.nih.gov/pmc/articles/PMC5920077/) that used a similar tube test protocol (as well as their review of the relevant literature).– Overall, I strongly suggest to avoid the definition of "dominant" and "subordinate" in the manuscript and to analyze their very nice data according to a more neutral "high rank" and "low rank" in the tube test (or even better to compare the 4 ranks individually, e.g. alpha, beta, etc) and try to determine, or at least speculate, of what that rank really means. As they and others have demonstrated, there is some very interesting biology associated with the rank assessed in this test. It clearly reflects an individual trait (similar to trait anxiety measured in the EPF for example?) that may or may not be related to social status, or social rank. Indeed, there is a vast literature showing that when social rank is assessed using behavioral observations of social interaction, subordinate mice manifested higher anxiety and depression-like behavior when compared to dominant mice, a finding which is opposite when ranks is assessed in the tube test (Figure 1).

There are several points in reviewer 2’s comments:

1) How the index of social dominance is measured. In order to explain this better, we have now added the following sentences to the corresponding part of the Materials and methods section:

“The designation of a winner after each of the 6 possible pairs of confrontations per cage allowed to rank each mouse by its winning times, varying between 0 – 3. […] In order to reflect the specificity of the observed dominant and subordinate behavior to the SCTT (Varholick et al., 2018), animals with highest index of dominance (ranks 1 and 2) are referred to as “high rank” herein, whilst animals with lowest index of dominance (ranks 3 and 4) are referred to as “low rank”.”

2) The validity of the Social Confrontation Tube Test to index individuals’ social hierarchy in the homecage. Although potential different explanations of the mice behavior in this test suggested by the reviewer are certainly valid, our experimental conditions have been chosen to minimize ambiguity in the interpretation of the data. Note that, as indicated by the reviewer, mice were housed together for at least 5 weeks before performing the tube test. This is highly important, as also indicated in one of the references provided by the reviewer (see above: Varholick JA, Bailoo JD, Palme R and Würbel H, Sci. Rep. 2018, Phenotypic variability between Social Dominance Ranks in laboratory mice – doi: 10.1038/s41598-018-24624-4), when they tested mice hierarchy dynamics throughout time from first establishment of the homecage colony, they “found that dominance ranks of most mice changed with time, but were most stable between the 2nd and 3rd week of testing”.

In our previous study (Larrieu et al., 2017) we also verified that the measured hierarchy was stable over time and established as well that it followed a linear pattern (see Author response image 1).

In addition, our choice for this experimental approach is based on a careful consideration of the validation of this test in the literature – see, for example, the following list of articles published on the methodological aspects or revision of the validity of the test throughout published data:

Fan Z, Zhu H, Zhou T, Wang S, Wu Y, Hu H. Using the tube test to measure social hierarchy in mice. Nat Protoc. 2019 Mar;14(3):819-831. doi: 10.1038/s41596-018-0116-4. PMID: 30770887.

Wang F, Kessels HW, Hu H. The mouse that roared: neural mechanisms of social hierarchy. Trends Neurosci. 2014 Nov;37(11):674-82. doi: 10.1016/j.tins.2014.07.005. Review. PMID: 25160682

3) The cross-validation of the results from the tube test with other confrontational tests. As done in (Wang et al., 2014), we also previously validated that co-habitation in tetrads in the homecage for corresponding changes in dominance behavior in other tests (e.g., agonistic/fighting behaviors and urine marking test) – see Author response image 1. In addition, the warmspot test was also recently used to validate the outcome of the tube test rank order outcome (see Zhou et al., 2017).

See Author response image 1 with the cross-validation of our current experimental conditions (at least 5 weeks of cohabitation in the homecage), as published in our study (Larrieu et al., 2017). When computing a dominance score including several submissive or offensive parameters (flight, avoidance, freezing, submissive posture; chasing attack including bite, upright and side offensive posture), we have shown that high rank (1-2) mice have a significantly higher dominance score as compared to low rank (3-4).

**Author response image 1. respfig1:** Hierarchical Rank Using a Social Confrontation Tube Test. (**A**) Illustration of the general timeline of the study. (**B**) Example of one cage representing the tube test ranks and winning times as a function of tube test trials. (**C**) Summary for nine cages over the 6-day test trials. (**D**) Time spent in the tube (**s**) as a function of the rank pairing (F5,48 = 9.78, p < 0.001, one-way ANOVA; ∗∗p < 0.01, ∗∗∗p < 0.001, Bonferroni’s test, n = 9 per rank pairing). (**E**) Dominance score after agonistic behaviors in the homecage (t28 = 2.30, ∗p < 0.05, unpaired t test, two-tailed n = 15 per group). (**F**) 2 × 2 contingency table for correlation between agonistic behaviors and tube test ranks (Fisher’s exact test, two-tailed, p = 0.050). (**G**) Left: picture representing typical urine marks profile of dominant and subordinate mice revealed by a UV light source. Right: 2 × 2 contingency table for correlation between urine marking test and tube test ranks (Fisher’s exact test, two-tailed, p = 0.026, n = 26 pairs). From Larrieu et al., 2017, Curr Biol..

4) The use of the terms ‘dominant’ and ‘subordinate’. As suggested by the reviewer, we have now modified all former references to these terms and substitute them, as suggested, by ‘high rank’ and ‘low rank’. Clarifications in the Materials and methods have been done as follows:

“In order to reflect the specificity of the observed dominant and subordinate behavior to the SCTT (Varholick et al., 2018), animals with highest index of dominance (ranks 1 and 2) are referred to as “high rank” herein, whilst animals with lowest index of dominance (ranks 3 and 4) are referred to as “low rank”.”

5) In addition, following the reviewer recommendation to discuss the study by Varholick et al., 2018, we have now added the following paragraph to the Discussion:

“However, a note of caution should be added as previous work suggests that whereas dominant mice tend to display more novelty-related exploratory behavior than less dominant mice, differences in anxiety-related behaviors seem to be less consistent (see Varholick et al., 2018, and references herein). We would also like to emphasize that we find a high reliability and linearity of social rank assessed under our experimental conditions that involve at least 5 weeks of cohabitation prior to carrying out the social confrontation tube test (see also Larrieu et al., 2017). This characterization may determine a different phenotype than studies in which social hierarchy is established within the first 2-3 weeks of cohabitation. Indeed, a recent study showed that during early cohabitation, dominance ranks of mice changed with repeated measurement, but became more stable between the 2nd and 3rd week of testing (Varholick et al., 2018).”

6) Finally, the reviewer asked how the results would look if we concentrated in a comparison between ranks 1 and 4. Comparison between rank 1 and 4, presented in Author response image 2, showing a similar trend as when comparing rank 1-2 with rank 3-4. Nevertheless, due to the strong reduction of animals per group using this approach (N=3), statistical interpretation needs to be cautious. As such, we have decided not to include these results in the manuscript. If the reviewers would like to see these results in the manuscript, we would of course reconsider our proposal.

**Author response image 2. respfig2:** Behavioral and metabolic comparisons between Rank 1 vs Rank 4. (**A**) Comparison of trait-anxiety parameters measured with an elevated plus maze when only the highest rank (R1) and lowest rank (R4) are compared. Student’s t-test, n=6-7 per group. (**B**) Comparison of trait-anxiety parameters measured with an open field when only the highest rank (R1) and lowest rank (R4) are compared. Student’s t-test, n=3 per group. (**C**) Depressive-like behavior measured as a composite z-score component of social avoidance and immobility time in animals of Rank 1 (F2,6=2.95, P>.05, one-way ANOVA; n=3 per group). (**D**) Depressive-like behavior measured as a composite z-score component of social avoidance and immobility time in animals of Rank 4 (F2,6=0.07, P>.05, one-way ANOVA; n=3 per group). (**E**) Accumbal neurochemistry in Rank 1 mice. One-way ANOVA, Bonferroni’s test, n=3 per group. (**F**) Accumbal neurochemistry in Rank 4 mice. One-way ANOVA, Bonferroni’s test, n=3 per group.

2) Figure 2. LAC treatment significantly improved stress-induced immobility (albeit the% change is modest) but had no effect in the social interaction test. Therefore, authors should be very careful in using strong statements on the "anti-depressant" effect of this drug.

We have now indeed softened the text according to the reviewer’s comments. Note, however, the we also found differences in the composite z-score for the two behavioral tests that indicates that, overall, CRS increases alterations in these emotional tests and LAC reverses it.

3) I see little added value in the use of the composite behavioral score of just 2 tests, out of which only one show a significant difference between groups. Essentially the score will be driven by the FST. Unless authors will be able to provide a strong rationale for its significance, this score should be removed.

Although we appreciate the reviewer’s comment, the composite score allows obtaining a more global figure of how individual animals are affected when tested a the two behavioral tests used. Therefore, this measurement has a higher significant value than regarding individual tests separately and, thus, unless this represents a major obstacle in the reviewer’s view, we would prefer to keep it in the article.

4) Figure 1F; please include actual total time in the interaction zone and the corners.

The time in interaction zone and in the corners has now been added in Figure 1F.

“Figure 1: (F) […] Time in interaction zone: Interaction: F_1,21_=12.80, P<.005; rank effect: F_1,21_=0.85, P<.05; stress effect: F_1,21_=12.21, P<.005, two-way ANOVA; p*<.05, **p<.005, Bonferroni’s test, n=6-7 per group / Time in corner zone: Interaction: F_1,21_=3.29, P>.05; rank effect: F_1,21_=0.28, P<.05; stress effect: F_1,21_=2.14, P>.05, two-way ANOVA, n=6-7 per group).”

5) Figure 1E; please include time in the center, time in the outer ring or corners etc. The significance of latency to enter the center as the most important measure of anxiety is questionable.

We have included the time in center as well as the distance travelled for the OF in the Figure 1E. These results have been described in the Results section and in the legend of Figure 1, as follows:

“However, the two groups did not show differences in the percent time spent in the center of the OF (Figure 1E, center), indicating that their difference in anxiety-like behaviors depends on the specific threat encountered. Importantly, no difference in locomotor activity was observed, as the distance travelled by both groups in the OF was similar (Figure 1E, right; n.s.).”

“Figure 1: (E) Anxiety-related behaviors measured in the open-field, including latency to first enter the center of the arena (*p<.05, unpaired t-test, two-tailed n = 14 per group) and time in center zone (n.s. unpaired t-test, two-tailed n = 14 per group). Locomotor activity is measured as the distance travelled in the OF (n.s. unpaired t-test, two-tailed n = 14 per group).”

Reviewer #3:The authors examined the potential antidepressant effects of Acetyl-L-Carnitine in dominant and subordinate mice exposed to chronic restraint stress. Dominant and subordinate mice were classified based on their hierarchical status. Following chronic stress, dominant mice show vulnerability in social interaction and in the forced swim test. Treatment with carnitine reversed the deficits seen in the Porsolt swim test but not social avoidance. The authors then used MRS and identified in the NAc some metabolites that are decreased by stress and rescued by carnitine. Some of these metabolites are related to mitochondrial function. Despite being correlational, this discovery approach has definitely some merits as it has identified some targets in the NAc that could be followed on with functional studies, in the hope that they play a major role in depression. While we liked this work, we still have a few major concerns that the authors need to address:1) The authors are relying too much on the forced swim test to demonstrate that L-Carnitine is an antidepressant drug. As the authors likely know, this behavioral test has received a lot of criticism lately and is definitely not a test for depression. Adding more behavioral tests -may be tests that require energy and coping- is necessary to strengthen this manuscript.

We thank the reviewer for this comment and, in fact, we use this test given its capability to distinguish between active and passive coping responses to a stressful situation. We take the point and will be aiming at increasing the battery of tests in our future studies.

2) The EPM data shown in Figure 1: Animals in this test spend 4-8% of their time in the open arms. This is an extremely low time that will not allow the experimenters to appreciate the effects of anxiogenic treatments.

Thanks for this comment. Given to normal inter-experiment variation (at least in our hands), percent time in open arms in this study was slightly (though non-significantly different) lower than results in other studies in the lab (see, e.g., our Larrieu et al., 2017, Curr Biol. Paper). As the purpose of this test in our study was to characterize basal anxiety-like behaviors in high and low rank animals, this point is not problematic regarding potential treatments, but we thank the reviewer for pointing out the issue and will aim at modifying the experimental conditions to make the test less anxiogenic whenever we aim at studying how a particular treatment may modify anxiety-like behaviors in mice.

3) OF data shown in Figure 1: why only latency to enter the center is reported? Usually, it is the time spent in the center of the OF that is reported

Thanks for this comment, we have now included these results in Figure 1E and reported the new data in the Results section.

4) Why rank 2 was considered dominant? If that is the case, rank 3 is also dominant over rank 4, right? Did the authors think about examining just the extremes (rank 1 vs. rank4)?

As mentioned earlier, given the low number of animals in our study, a comparison between only rank 1 vs. rank 4 was challenging. We have tested whether a significant difference existed between rank 1 and rank 4 in the different parameters of the basal anxiety measurements (OF and EPM) and post-CRS (depressive-like behavior and accumbal metabolites). While the tendency was similar as when including rank 1-2 vs. rank 3-4, the statistical test (Student’s t-test or one-way ANOVA with post-hoc) did not pass the threshold of significance, due to the reduced statistical power, but are reassuring that our grouping of animals in high and low rank animals reflect the changes observed in ranks 1 vs 4, as follows:

[Editors' note: further revisions were suggested prior to acceptance, as described below.]

The manuscript has been improved but there are some remaining issues that need to be addressed before acceptance, as outlined below.1) While the authors changed the nomenclature from dom/sub to high/low rank in the Result section/figures, the interpretation of what that rank reflects is unchanged in the Abstract, Introduction and large part of the Discussion. As currently written, the role of status/rank in the observed phenotype is confusing. If authors consider social rank a trait (genetically encoded?) rather than a state, they should be consistent through the manuscript.

We have modified this in the text:

Abstract: we include now the following sentence and wording:

“High rank, but not low rank, mice, as assessed with the tube test,….”

Results: we include now the following wording:

the behavioral phenotype of high and low rank mice “(as assessed by testing the 4 mice from the same home cage in the social confrontation tube test),”

We have also added “high (ranks 1 and 2) and low (ranks 3 and 4) rank mice”

In the Discussion we have now included the following sentence:

“High rank animals were defined as those that following cohabitation in groups of 4 males, emerged as ranks 1 and 2 in the social confrontation tube test.”

2) Authors did not address a main concern: high and low rank groups simply reflect a median split of the population. A visual inspection of Figure 1 clarifies that ranks 2 and 3 are very similar, essentially indistinguishable from each other. Because the N is low, they could not analyze each rank separately, and rank 1 vs rank 4 are not statistically different (Author response images).

We would like to indicate that the figures that we included in the revision R1 rebuttal are supportive of differences in results between rank 1 and rank 4. It should be noted that in our report, we applied two-tailed statistics; as the comparisons were made following specific predictions, one-tailed statistics for Rank 1 animals in those results give trends very close to significance (i.e., p<0.056, P<0.075, p<0.08; p<0.05; p<0.1; p<0.12; p<0.056; p<0.056) while they are clearly not significant for Rank 4 mice, further supporting the reported results.

In order to address this issue from the reviewer more explicitly, we have now added the following note of caution to the Discussion section:

“An additional issue to consider is that, for our analyses, we have grouped ranks 1 and 2 as high rank mice and ranks 3 and 4 as low rank. Although this grouping allows revealing statistical differences in the evaluated variables, each of the ranks in a tetrad home cage hierarchy may in fact lead to idiosyncratic phenotypes. In the future, it would be important to address vulnerability to stress for each specific rank in the colony.”

3) The authors did not provide any justification for the use of the composite behavioral score (except for convenience of having a second measure of "global" depression-like behavior).

In fact, in our rebuttal to R1, we provided an explanation for the use of the composite behavior, as follows:

“Although we appreciate the reviewer’s comment, the composite score allows obtaining a more global figure of how individual animals are affected when tested across the two behavioral tests used. Therefore, this measurement has a higher significant value than regarding individual tests separately and, thus, unless this represents a major obstacle in the reviewer’s view, we would prefer to keep it in the article.”

In the manuscript we had also included the following explanation:

“In order to obtain an integrated estimation of how stress and LAC treatments affect emotional behavior more globally, we computed an overall behavioral composite to integrate deviation from normality considering high rank mice variance in both behavioral tests.”

Perhaps this was not clear enough and, therefore, we have now added the following additional explanation to the text:

“Specifically, the use of this composite score allows considering variation from the mean in individuals’ behavior across two different behavioral tests, providing a more robust measurement of individuals’ behavior in tests that are typically used to index mice depressive-like behaviors.”

In other words, by computing animal’s behavior (or their deviation from normality/mean) across behavioral tests, we allow revealing consistent changes across tests, which in fact is a different measurement that computing the different behaviors separately; something that do not allow revealing whether the same animals were similarly or differently affected across tests; this composite behavior allows a more global picture of individuals’ deviation in global behavioral testing.

We hope this additional explanation in the manuscript and here in this rebuttal R2 helps us making a strong case for why we believe that this is an important additional measurement to add to the manuscript.

4) Finally, the evidence in favor of high rank mice being characterized by high anxiety is limited to only 1 behavioral test, the EPM.

Yes, this is indicated in the text, already from our revision in R1, as follows: “indicating that their difference in anxiety-like behaviors depends on the specific threat encountered”.

In order to make this point now clearer, we have added the following explanation in the Discussion:

“….higher rank is related to higher anxiety-like behaviors as indicated by their exploration of an elevated plus maze (but note that these animals did not differ from low rank in the time spent in the center of the open field, only in their latency to enter the zone)”

I would like to ask the authors to address these comments by thoroughly discussing the strengths/limits of the ranking dominance classifications.

We have tried now to make all those revisions more explicit and hope the manuscript is now in a satisfactory form (see specific explanations above).